# Shorebirds-driven trophic cascade helps restore coastal wetland multifunctionality

Chunming Li[1], Jianshe Chen [1], Xiaolin Liao[2], Aaron P. Ramus[3], Christine Angelini[4], Lingli Liu [5], Brian R. Silliman[6], Mark D. Bertness[7] & Qiang He [1] ✉

Ecosystem restoration has traditionally focused on re-establishing vegetation and other foundation species at basal trophic levels, with mixed outcomes. Here, we show that threatened shorebirds could be important to restoring coastal wetland multifunctionality. We carried out surveys and manipulative field experiments in a region along the Yellow Sea affected by the invasive cordgrass *Spartina alterniflora*. We found that planting native plants alone failed to restore wetland multifunctionality in a field restoration experiment. Shorebird exclusion weakened wetland multifunctionality, whereas mimicking higher predation before shorebird population declines by excluding their key prey – crab grazers – enhanced wetland multifunctionality. The mechanism underlying these effects is a simple trophic cascade, whereby shorebirds control crab grazers that otherwise suppress native vegetation recovery and destabilize sediments (via bioturbation). Our findings suggest that harnessing the top-down effects of shorebirds – through habitat conservation, rewilding, or temporary simulation of consumptive or non-consumptive effects – should be explored as a nature-based solution to restoring the multifunctionality of degraded coastal wetlands.

To achieve sustainable development goals, the United Nations has designated 2021–2030 as the Decade on Ecosystem Restoration, and countries worldwide are increasingly committed to restoring degraded ecosystems and regenerating the services natural ecosystems provide to humanity[1]. The Kunming-Montreal Global Biodiversity Framework recently agreed upon by 196 countries proposed a goal to restore at least 30% of degraded ecosystems by 2030. To fulfill these ambitious goals, governments, non-governmental organizations, and local communities need effective, scalable approaches to restoring ecosystems. While society values ecosystems for the simultaneous provision of multiple functions (i.e., multifunctionality[2–5]) such as food provision, pollution mitigation, and carbon sequestration, how to restore ecosystem multifunctionality remains poorly understood. Uncertainty in our ability to restore ecosystem multifunctionality is constraining support from different stakeholders and, consequently, the human capital and financial investments required to meet international ecosystem restoration goals.

Current restoration efforts typically focus on re-establishing vegetation or other foundation species like oysters and corals at basal trophic levels. This approach, the Field of Dreams hypothesis[6], assumes that the recovery of foundation species will lead to the recovery of higher trophic levels and ecosystem functions through

[1]Coastal Ecology Lab, MOE Key Laboratory for Biodiversity Science and Ecological Engineering, National Observations and Research Station for Wetland Ecosystems of the Yangtze Estuary, School of Life Sciences, Fudan University, 2005 Songhu Road, Shanghai 200438, China. [2]College of Ecology and Environment, Nanjing Forestry University, Nanjing, Jiangsu 210037, China. [3]Department of Biology and Marine Biology, University of North Carolina Wilmington, Wilmington, NC, USA. [4]Department of Environmental Engineering Sciences, University of Florida, Gainesville, FL, USA. [5]State Key Laboratory of Vegetation and Environmental Change, Institute of Botany, Chinese Academy of Sciences, Xiangshan, Beijing 100093, China. [6]Nicholas School of the Environment, Duke University, 135 Duke Marine Lab Road, Beaufort, NC 28516, USA. [7]Department of Ecology, Evolution and Organismal Biology, Brown University, Providence, RI 02912, USA. ✉e-mail: he_qiang@hotmail.com

bottom-up processes such as provision of nutritional resource and habitat structure. Recovery of foundation species and ecosystem functions, however, is often slow or incomplete, even decades after restoration[7–9]. Although human-driven loss of wildlife at higher trophic levels has been widely documented to disrupt ecosystem functions via trophic interactions[10], restoration projects often neglect the potential top-down effects of wildlife at higher trophic levels as an integral component. It has also been a concern that wildlife at higher trophic levels can either increase or decrease ecosystem functions[11–13], but how they affect the restoration of ecosystem multifunctionality remains largely unknown.

Here, we assessed the role of threatened shorebirds (i.e., Charadriiformes) in restoring the multifunctionality of coastal wetlands, one of the most valuable human service-providing ecosystems on Earth. Coastal wetlands around the world generate vital ecosystem functions, such as carbon sequestration, pollution mitigation, and wave dissipation[14]. In the Yellow Sea where this study is focused, these systems also provide critical stopover habitats for once abundant shorebirds that migrate along the East Asian-Australasian Flyway[15–17]. Coastal wetlands in the Yellow Sea, however, have been severely degraded[18,19]. Even in protected areas where anthropogenic activities are reduced, invasion of the exotic smooth cordgrass *Spartina alterniflora* has transformed marshes formerly dominated by native plants including *Scirpus mariqueter*[18]. The shift towards dense, tall stands of invasive *Spartina* has diminished the value of stopover habitat for shorebirds

that are adapted to forage for invertebrate prey, such as grazing crabs, within the short and loose structure of native marsh plants, contributing to shorebird population declines[20]. Our regional-scale synthesis revealed that (i) shorebird abundance declined at 10 of the 14 surveyed sites (by 33–73%; Fig. 1a) and for 32 of the 45 recorded species (by 13–99%; Fig. 1b) from ~2000 to the early 2010s in the Yellow Sea[21] and (ii) 68% of the 60 shorebird species that utilize coastal habitats in the Yellow Sea have shrinking global populations (Fig. 1b; also see ref. 16). These declining trends of shorebird populations were consistently observed for species that consume and do not consume crabs (Supplementary Fig. 1). Population declines are indeed drastic for birds globally[18,22,23]. Loss of shorebirds may disrupt a prevalent tritrophic interaction that maintains native wetlands in this region. Our synthesis revealed that shorebirds (see Supplementary Table 1 for a list of crab predator species) readily consume grazing crabs, which are the primary generalist herbivores that strongly suppress native plants in these marshes (Fig. 1d). The implications of this tritrophic interaction for restoring the multifunctionality of these coastal wetlands could be profound but have yet to be tested.

We hypothesized that following exotic cordgrass eradication, shorebirds are critical for rebuilding wetland multifunctionality due to a tritrophic interaction, in which shorebirds control grazing crab populations that can impact wetland functions directly through bioturbation and indirectly by regulating the recovery of native vegetation. To test this hypothesis, we initiated a restoration experiment (the

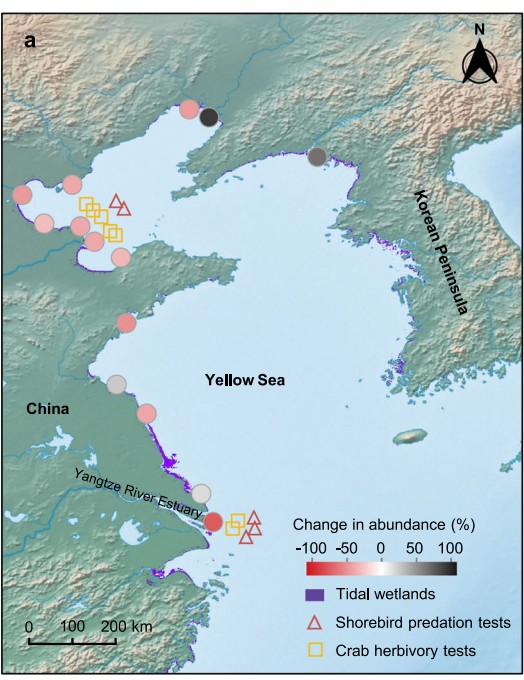
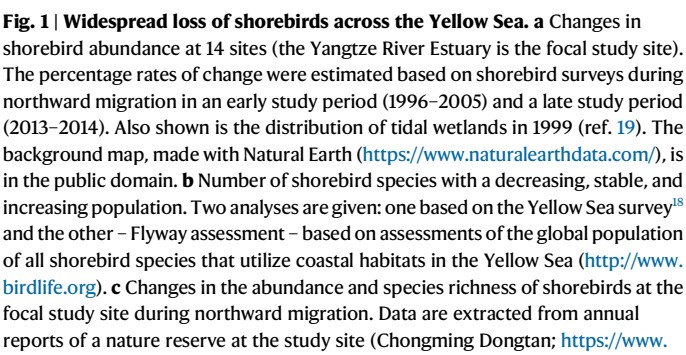
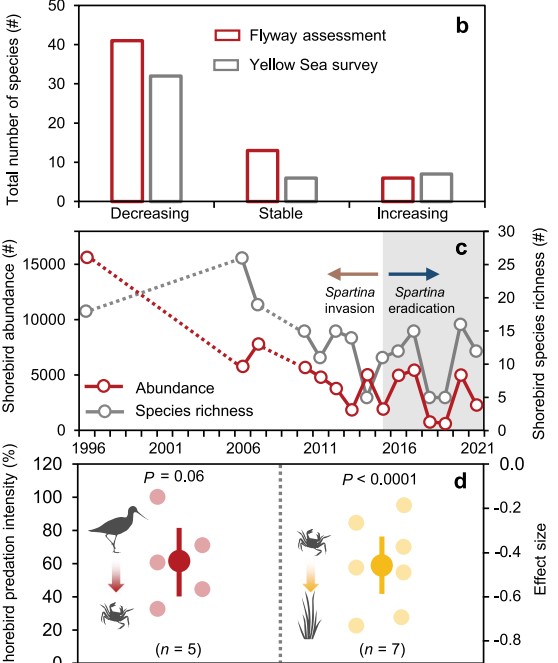

**Fig. 1 | Widespread loss of shorebirds across the Yellow Sea. a** Changes in shorebird abundance at 14 sites (the Yangtze River Estuary is the focal study site). The percentage rates of change were estimated based on shorebird surveys during northward migration in an early study period (1996–2005) and a late study period (2013–2014). Also shown is the distribution of tidal wetlands in 1999 (ref. 19). The background map, made with Natural Earth (https://www.naturalearthdata.com/), is in the public domain. **b** Number of shorebird species with a decreasing, stable, and increasing population. Two analyses are given: one based on the Yellow Sea survey[18] and the other – Flyway assessment – based on assessments of the global population of all shorebird species that utilize coastal habitats in the Yellow Sea (http://www.birdlife.org). **c** Changes in the abundance and species richness of shorebirds at the focal study site during northward migration. Data are extracted from annual reports of a nature reserve at the study site (Chongming Dongtan; https://www.

dongtan.cn/). Exotic *Spartina alterniflora* was eradicated since 2015 (shaded area). **d** Meta-analysis of shorebird predation intensity (left) and crab grazing effects on native plants (right) at different sites (see panel **a** for locations). Data are shown as means with error bars for 95% confidence interval. *n* indicates the number of independent sites (for shorebird predation intensity) or tests (for crab grazing effects). Shorebird predation intensity indicates the percentage of tethered crabs eaten by shorebirds within 24 h and should be interpreted mainly as the presence, rather than the absolute natural rate, of shorebird predation. Effect sizes are log response ratios, with negative ones indicating that crabs suppress plant biomass. Data of shorebird predation intensity were tested against zero using a one-sample nonparametric Wilcoxon test (two-sided), and effect sizes of crab grazing were tested using a random-effects model. See "Methods" for further details.

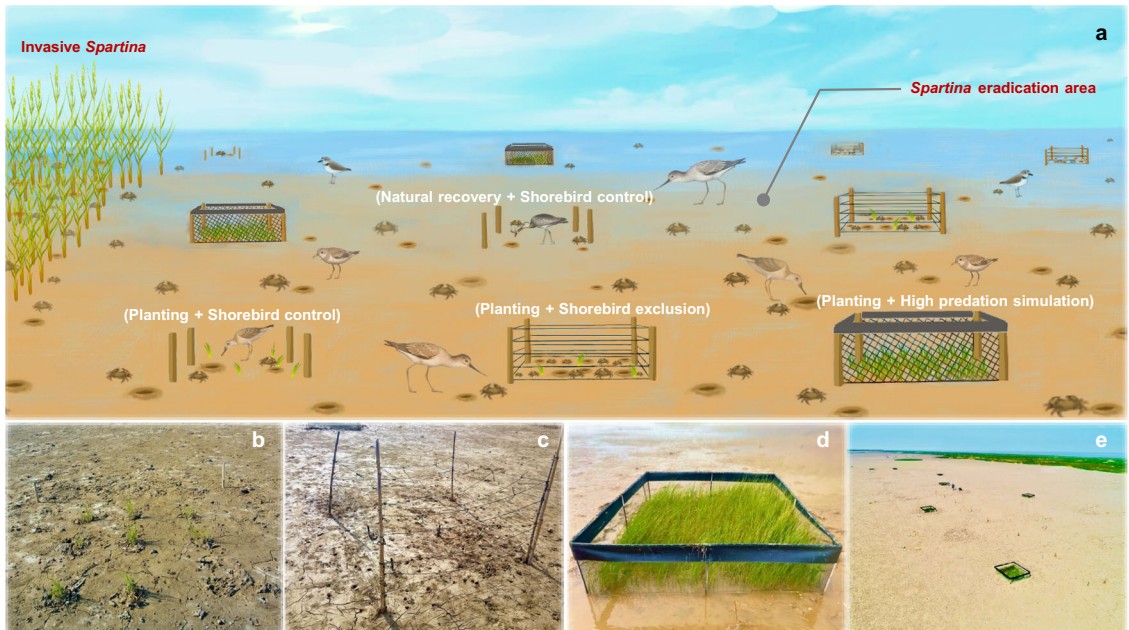

**Fig. 2 | The Coastal Wetland Trophic Restoration experiment. a** An illustration of the experimental design. The four experimental treatments (natural recovery + shorebird control, planting + shorebird control, planting + shorebird exclusion, and planting + high predation simulation) were replicated 8 times in 4 m² plots in a site where invasive plants (*Spartina alterniflora*) were successfully eradicated since 2015 and native plants had not recovered naturally. **b**–**d** Photographs of planting + shorebird control (**b**), planting + shorebird exclusion (**c**), and planting + high predation simulation treatments (**d**). **e** Bird's-eye view of the experiment. Credits: Chunming Li (illustration) and Qiang He (all photographs).

Coastal Wetland Trophic Restoration experiment) in 2018 in a coastal salt marsh in the Yangtze estuary, where cordgrass invasion began in the mid-1990s and, since then, shorebird abundance (Fig. 1c) and density (Supplementary Fig. 2) have declined by >80%. This trend of decline held even if the 1996 shorebird population data was an overestimate of the historical baseline (see "Methods") and was likely conservative, with an annual rate of decline (−3.4%) lower than previous estimates for shorebirds with a strong reliance on the Yellow Sea tidal wetlands (−5.2% on average; see ref. 16). Although exotic cordgrass was successfully eradicated in 2015, neither shorebird populations nor native vegetation have recovered (Fig. 1c). Our restoration experiment had four treatments (Fig. 2): (i) natural recovery + shorebird control: control with no additional treatments to allow the spontaneous recovery of native vegetation; (ii) planting + shorebird control: *Scirpus mariqueter*, a dominant endemic clonal sedge, was planted using standard restoration practices to enhance native vegetation recovery; (iii) planting + shorebird exclusion: prior to planting *Scirpus*, we excluded shorebirds to simulate restoration under scenarios of exacerbated shorebird population loss; and (iv) planting + high predation simulation: prior to planting *Scirpus*, we excluded crab grazers nearly completely to simulate historically higher levels of shorebird predation. This simulation was warranted, given that there are multiple lines of experimental and observational evidence that high abundances of shorebirds could drive crab grazers to become rare or locally absent, consistent with many previous studies on the impacts of predators on prey populations[24–32] (see further discussions in "Methods"). Then, we monitored the abundance of shorebirds, crabs, and *Scirpus* monthly for three years (2019–2021) and, in the third year, quantified twelve wetland functions that underpin services for which coastal wetlands are highly valued[14] (Supplementary Table 2). We generally followed previous studies[4,33,34] to quantify a variety of important biological (primary production, secondary production of macrofauna, and microbial production), physical (wave dissipation, marsh infiltration, and sediment accretion), and biogeochemical (soil respiration, nitrogen mineralization, litter decomposition, sediment carbon burial, nitrogen accumulation, and soil heavy

metal reduction) functions, covering both above- and belowground. We then assessed the impacts of restoration treatments on these wetland functions individually and collectively. We found that planting alone failed to restore native vegetation or enhance any ecosystem functions and that shorebird exclusion weakened wetland multifunctionality, whereas mimicking higher predation before shorebird population declines enhanced wetland multifunctionality. These results highlight that harnessing the top-down effects of shorebirds may potentially provide a nature-based solution to restoring the multifunctionality of coastal wetlands.

## Results and discussion
### Natural recovery and planting failed for restoration
We found that natural recovery and planting without regard to shorebirds or their key prey – crab grazers – were ineffective for restoring vegetation and the delivery of ecosystem functions in these coastal wetlands. Natural recovery of native vegetation had not occurred in the seven years following cordgrass eradication. Planted *Scirpus* was also unable to become established because it was quickly consumed by crab grazers during the first few months following planting (Fig. 3b, d, e). Crabs make burrows, which could affect sediment physical conditions[35]. However, most planted *Scirpus* clumps, including those not adjacent to crab burrows, were quickly grazed by crabs, confirming crab grazing as the key mechanism by which crabs control *Scirpus* (see Supplementary Fig. 3 and Supplementary Movie 1; ref. 36). As a result, planting did not significantly affect any of the 12 wetland functions we measured in comparison to natural recovery treatments (Fig. 4).

Restoration of coastal wetlands in the Yellow Sea, as well as globally, is often implemented by removing invasive species or other types of disturbances and allowing wetlands to recover naturally, or by planting or seeding to accelerate the recovery process, often at great cost[37]. These restoration approaches may be effective at sites with no or low levels of grazing pressure (e.g., ref. 38). Our results, however, suggest that they would be ineffective at sites with strong grazing pressure. This finding is supported by recent studies in other

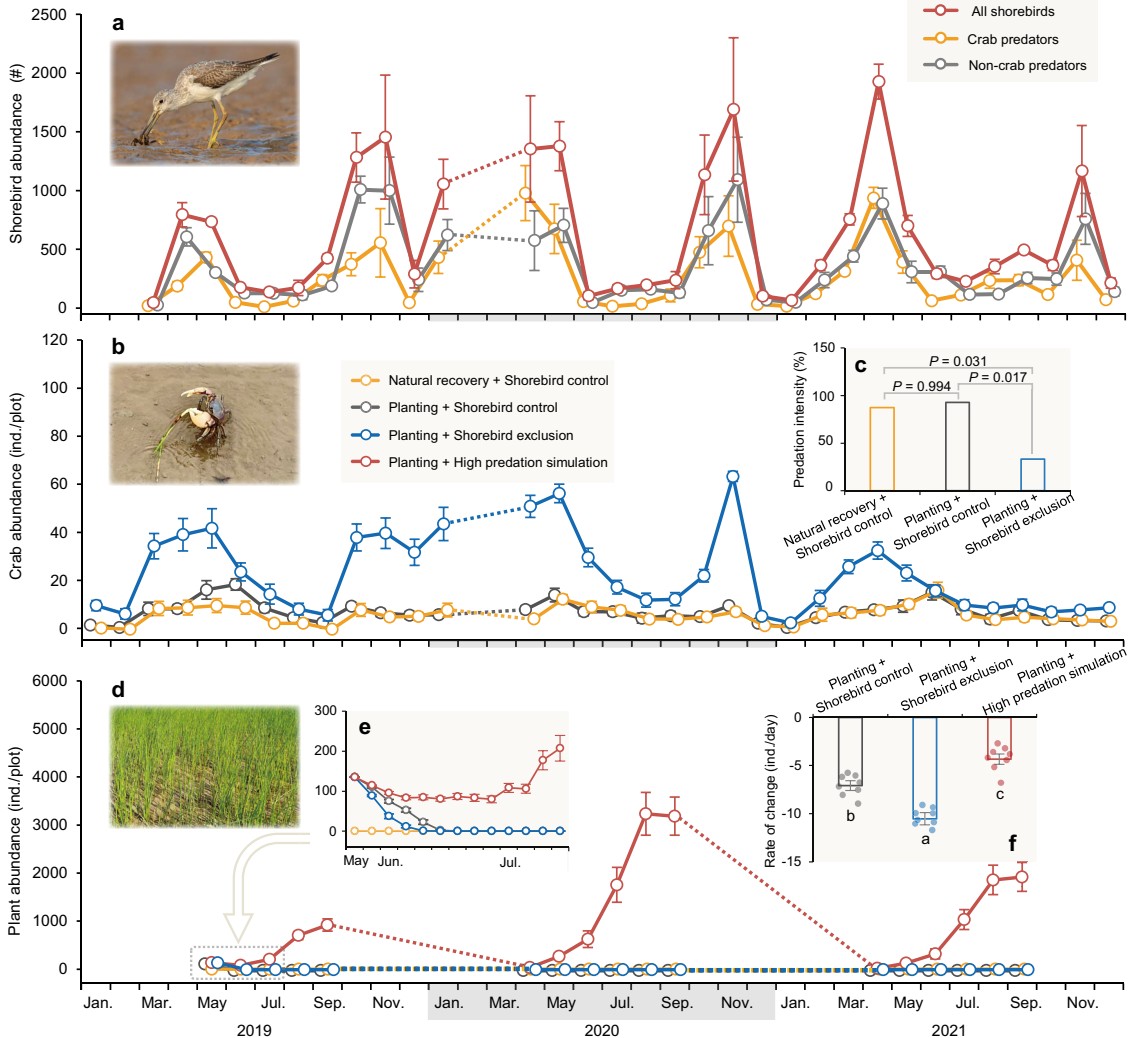

**Fig. 3 | Shorebird, grazer, and plant abundances from 2019 to 2021 in the Coastal Wetland Trophic Restoration experiment.** **a** Total shorebird abundance and those of species that consume or do not consume crabs. Data are shown as means with error bars for standard errors (*n* = 3 independent camera traps). **b** Crab grazer abundance. Data are shown as means with error bars for standard errors (*n* = 8 independent plots per treatment). **c** Shorebird predation intensity on tethered grazers (a pairwise proportion test, two-sided, with *P* values adjusted using the Bonferroni method). **d** Plant abundance (*Scirpus* is a perennial and overwinters via belowground rhizomes and corms) (*n* = 8 independent plots per treatment). **e** Close-up of plant abundance from May to July 2019 (*n* = 8 independent plots per

treatment). **f** Rate of plant abundance change over the first week following planting (no plants in natural recovery + shorebird control treatments throughout). Bars that do not share a letter differ significantly from one another based on Tukey's HSD multiple comparisons following a one-way ANOVA (*P* < 0.05; df = 2, 21, *F* = 59.81, *P* < 0.0001; see Supplementary Table 6 for detailed test statistics). Data are shown as means with error bars for standard errors (*n* = 8 independent plots per treatment). Dashed lines indicate periods without observation (due to COVID lockdown or non-plant growing season). Photo credits: Shuyan Zhang (shorebird) and Chunming Li (crab and *Scirpus*).

ecosystems showing that active restoration, often implemented through planting and preferred by policymakers and restoration practitioners[39], did not consistently enhance the recovery of ecosystems in comparison to natural regeneration[40,41]. In our study, some crabs nearby might have been attracted to plantings in small plots, contributing to the observed effects of crab grazing. Nonetheless, crabs still intensively constrained the establishment of marsh plantings at real-world restoration projects conducted at our study site and other sites in the Yellow Sea[42]. This is consistent with a recent global synthesis which found that the negative effects of herbivores on plant abundance at restoration sites often did not vary significantly with plot size and remained strong in restoration studies that used > 1 ha plots[41], which are larger than the sizes of many restoration sites[37].

### Exacerbation of ongoing shorebird loss impeded restoration

The success of restoration efforts may diminish if already depleted shorebird populations continue to decline. We found that exclusion of

the relatively few remaining shorebirds consistently triggered seasonal outbreaks of crab grazers each year and significantly decreased several wetland functions. Shorebird exclusion significantly decreased predation on crab grazers (Fig. 3c) and led to increased grazer abundances, an effect that varied strongly with time (generalized linear mixed model: Treatment: df = 2, $\chi^2$ = 45.60, *P* < 0.0001; Time: df = 33, $\chi^2$ = 415.44, *P* < 0.0001; Treatment × Time: df = 66, $\chi^2$ = 481.08, *P* < 0.0001). Shorebird exclusion more strongly increased grazer abundances in spring and autumn when migratory shorebirds are most abundant at the restoration site (Fig. 3a, b, Supplementary Fig. 4). Indeed, the effect size of shorebird exclusion on grazer abundance (i.e., the natural log of grazer abundance in planting + shorebird exclusion treatments divided by that in planting + shorebird control treatments) amplified with increasing abundance of shorebirds (Supplementary Fig. 5, Supplementary Table 3). This trend was (i) driven primarily by shorebirds that accounted for 86% of all birds we observed, including *Charadrius alexandrines*, *Calidris alpina*, *Calidris*

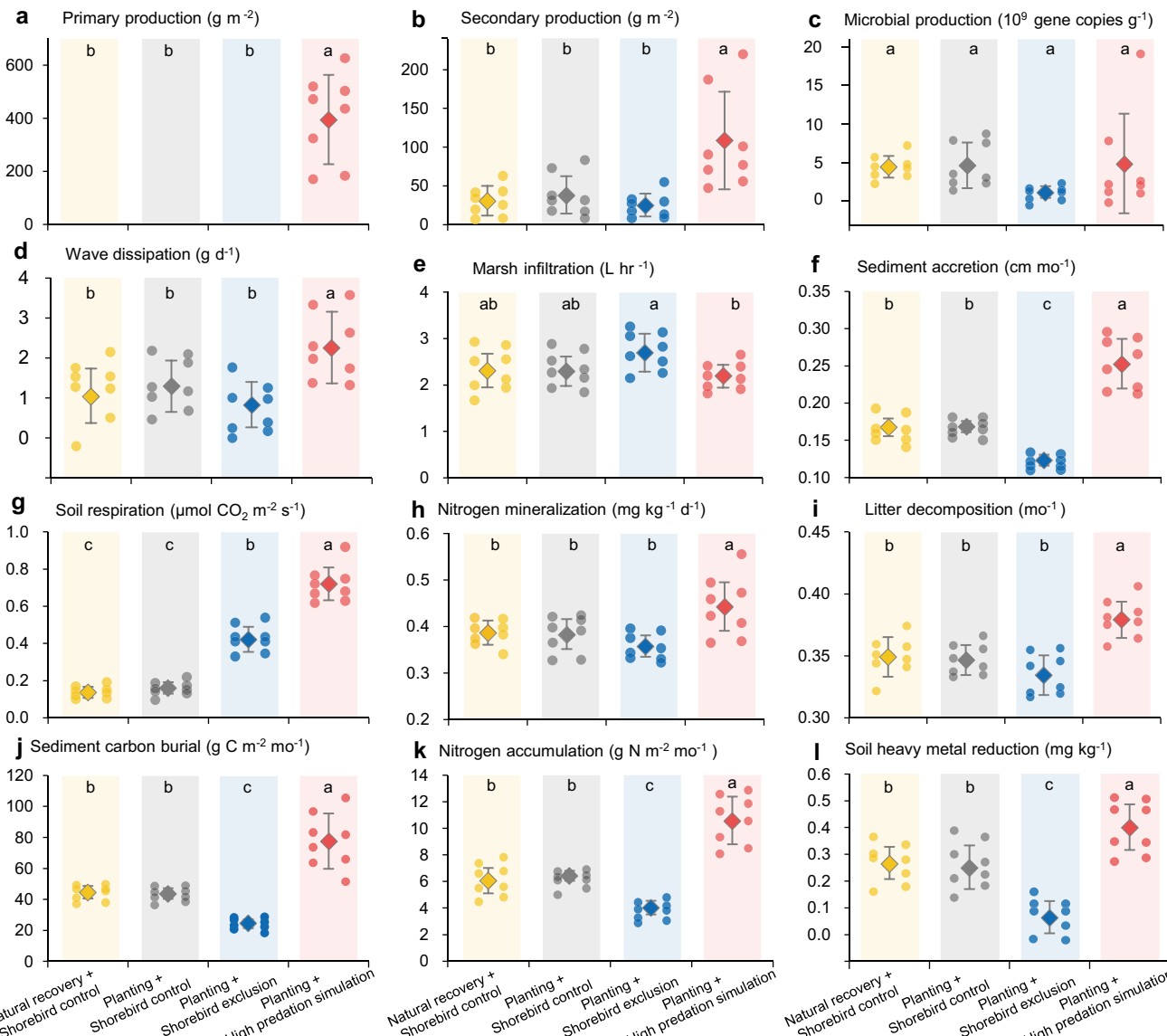

**Fig. 4 | Wetland functions in different restoration treatments. a** Primary production, **b** secondary production of macrofauna, **c** microbial production, **d** wave dissipation (estimated by the rate of gypsum block dissolution given in Supplementary Fig. 16; see "Methods"), **e** marsh infiltration, **f** sediment accretion, **g** soil respiration, **h** nitrogen mineralization, **i** litter decomposition, **j** sediment carbon burial, **k** nitrogen accumulation, and **l** soil heavy metal reduction (estimated with the Nemerow multifactor index given in Supplementary Fig. 16; see "Methods"). For all wetland functions, higher values indicate greater ecosystem service benefits (Supplementary Table 2). For primary production and secondary production, bars that do not share a letter differ significantly from one another based on pairwise comparisons following a nonparametric Wilcoxon test ($P < 0.05$, adjusted with the bonferroni method; see Supplementary Tables 7 & 8 for detailed test statistics). For the other functions, bars that do not share a letter differ significantly from one another based on Tukey's HSD multiple comparisons following a one-way ANOVA ($P < 0.05$; see Supplementary Tables 7 & 8 for detailed test statistics). Data are shown as means with error bars for standard errors ($n = 8$ independent plots per treatment). Data points of the same treatment had the same color and had shadings of a similar color.

*tenuirostris*, *Numenius phaeopus*, and *Xenus cinereus* (Supplementary Fig. 6), (ii) irrelevant to non-shorebird species (including herons, egrets, and gulls) that were uncommon at our study site, particularly during seasons when crabs were most abundant, regardless of whether they consume crabs (Supplementary Figs. 7 and 8), and (iii) irrelevant to overall bird species richness (Supplementary Fig. 9, Supplementary Table 3). This trend was evident not only for shorebird species that consume crabs, but also for those that do not consume crabs (Supplementary Fig. 5, Supplementary Table 3), likely due to non-consumptive effects of shorebirds. Indeed, in a supplementary experiment, we found that the presence of a swinging shorebird mimic reduced the abundance of crab grazers to almost nil over just a few weeks (Supplementary Fig. 10, Supplementary Tables 4 and 5). Without the top-down consumptive and non-consumptive effects of

shorebirds on crab grazers, shorebird exclusion accelerated the grazing-driven loss of planted *Scirpus* by 49% and precluded vegetation recovery (Fig. 3e, f; Supplementary Table 6).

Shorebird exclusion significantly decreased four of the 12 examined wetland functions: sediment accretion by 27% (Fig. 4f), sediment carbon burial by 42% (Fig. 4j), nitrogen accumulation by 37% (Fig. 4k), and soil heavy metal reduction by 74% (Fig. 4l) in comparison to planting + shorebird control treatments (Supplementary Tables 7 and 8). These wetland functions decreased in shorebird exclusion treatments most likely as a result of increased sediment bioturbation by burrowing crabs. This is supported by the significantly negative relationships between all these four wetland functions and crab abundance (Supplementary Fig. 11, Supplementary Table 9) and by previous observational and experimental studies showing that high crab

densities contribute to high sediment erosion[43] and carbon emission[44]. Although shorebird exclusion enhanced soil respiration by 161% (Fig. 4g), also likely by increasing crab bioturbation (Supplementary Fig. 11f), it did not affect the other seven wetland functions, including primary production, secondary production, microbial production, marsh infiltration, wave dissipation, nitrogen mineralization, and litter decomposition (Fig. 4). These effects were similar when planting + shorebird exclusion treatments were compared with natural recovery + shorebird control treatments (Fig. 4).

## Simulating higher shorebird predation enhanced restoration

In contrast, experimentally simulating historically higher levels of shorebird predation by excluding their key prey – crab grazers – enabled the establishment of planted *Scirpus*, led to the recovery of native vegetation (generalized linear mixed model: Treatment: df = 3, $\chi^2$ = 182.66, $P$ < 0.0001; Time: df = 16, $\chi^2$ = 2771.75, $P$ < 0.0001; Treatment × Time: df = 48, $\chi^2$ = 7984.72, $P$ < 0.0001), and significantly enhanced many wetland functions. In higher shorebird predation simulation treatments, the density of planted native vegetation increased by >700% within only a few months in the first growing season and further increased by 200-300% over the following two years (Fig. 3d–f). Compared to planting alone, simulating high predation facilitated primary production (Fig. 4a), and enhanced secondary production by 183% (Fig. 4b), wave dissipation by 75% (Fig. 4d), sediment accretion by 50% (Fig. 4f), soil respiration by 347% (Fig. 4g), nitrogen mineralization by 15%, litter decomposition by 9% (Fig. 4h), sediment carbon burial by 73% (Fig. 4j), nitrogen accumulation by 64%, and soil heavy metal reduction by 59% (Fig. 4l) (Supplementary Tables 7 and 8). Simulating high predation did not significantly affect microbial production and marsh infiltration (Fig. 4c, e). These effects were similar when planting + high predation simulation was compared to natural recovery + shorebird control treatments and became even manifest when compared to planting + shorebird exclusion treatments (Fig. 4, Supplementary Table 8). We found that all wetland functions (except microbial production and marsh infiltration), including functions unaffected by crab abundance (Supplementary Fig. 11, Supplementary Table 9), had a significant, positive relationship with primary productivity (Supplementary Fig. 12, Supplementary Table 10). These results highlight the profound importance of vegetation recovery, a process precluded or slowed by intense grazing in many coastal wetlands of the Yellow Sea[36,45] and, more broadly, other predator-depleted ecosystems globally[46–48].

## Strong restoration effects on wetland multifunctionality

Our assessment of multifunctionality using an average standardized index of the 12 wetland functions showed that without regard to shorebirds, planting did not affect wetland multifunctionality compared to natural recovery treatments (Fig. 5a, Supplementary Table 11). Compared to planting + shorebird control treatments, excluding shorebirds reduced multifunctionality by 20%, while simulating high predation increased multifunctionality by 61% (Fig. 5a). These effects were generally consistent when multifunctionality was assessed by using the effective number of functions - the equivalent number of functions were all functions provided equally (Supplementary Fig. 13, Supplementary Tables 12 and 13), and using effective multifunctionality - a measure of the cumulative performance of the system that takes into account both the effective number of functions and the average level at which the functions are performed (Fig. 5b, Supplementary Tables 14 and 15). Separate analyses of above- versus belowground multifunctionality further revealed that excluding shorebirds primarily affected belowground multifunctionality, while simulating high predation significantly affected both above- and belowground multifunctionality (Supplementary Fig. 14, Supplementary Table 16). Although planting and simulating high predation enabled the establishment and recovery of native vegetation, the increase in

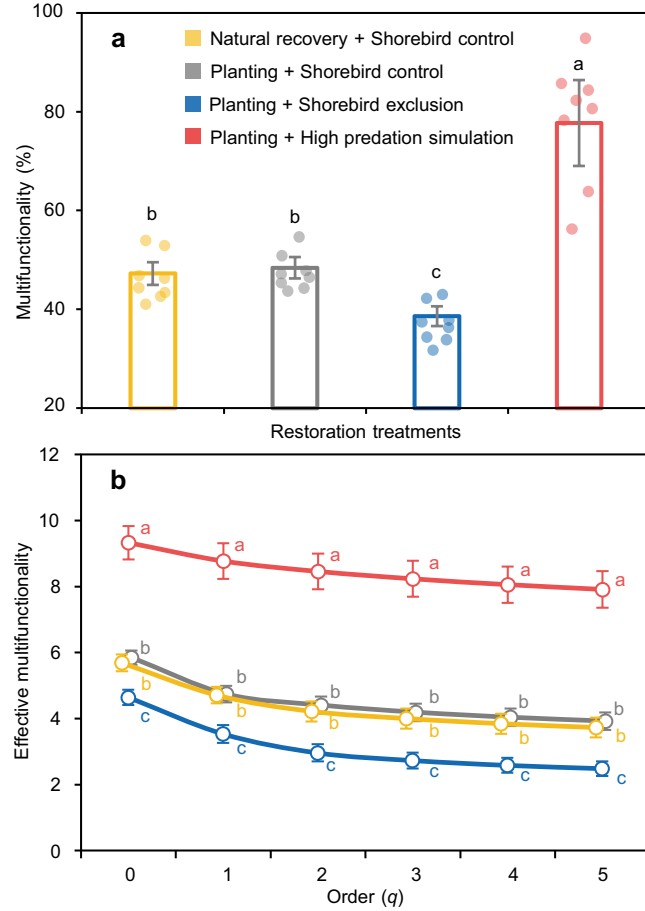

**Fig. 5 | Wetland multifunctionality in different restoration treatments.** **a** Averaged multifunctionality index of the twelve wetland functions quantified. Treatments that do not share a letter differ significantly from one another based on Tukey's HSD multiple comparisons following a one-way ANOVA ($P$ < 0.05; df = 3, 28, $F$ = 54.25, $P$ < 0.0001; see Supplementary Table 11 for detailed test statistics). **b** The effective multifunctionality in different restoration treatments. Effective multifunctionality is a measure of the cumulative performance of the system were all functions provided equally and takes into account both effective number of functions (see Supplementary Fig. 13) and average level of functions[89]. The effective number of functions is the equivalent number of functions were all functions provided at the same level. The effective number of functions is just the number of functions measured when order $q = 0$ and analog to Shannon diversity for species when $q = 1$. When $q > 1$, functions performing at higher levels are given greater weight (see "Methods"). Data are shown as means with error bars for standard errors ($n = 8$ independent plots per treatment). Treatments that do not share a letter differ significantly from one another based on Tukey's HSD multiple comparisons following a one-way ANOVA ($P$ < 0.05; see Supplementary Tables 14 and 15 for detailed test statistics).

multifunctionality we observed cannot be fully attributed to the recovery of native vegetation alone. Correlations between primary production and individual ecosystem functions were often of moderate strength (all $|r|$ < 0.8) and varied from positive to negative (Supplementary Fig. 15).

These findings provide unique evidence that shorebirds can help restore the multifunctionality of a coastal wetland following eradication of invasive cordgrass. Our results reveal that the top-down effects of shorebirds and their prey – crab grazers extend beyond biotic functions such as productivity and cascade to influence a broad suite of physical and biogeochemical functions, both above- and belowground. Our results based on a three-year experiment are likely conservative, as the effects of shorebirds could become more pronounced over a longer period of time, particularly for slow ecosystem functions

that are amplified through long-term biogeographic feedback, such as sediment carbon burial. Furthermore, the results of our crab exclusion treatments should be interpreted with caution as simulation of the historical effects of shorebirds, given potential uncertainties in the historical impacts of shorebirds on crab grazers. However, even if shorebirds of historically high abundances did not nearly eliminate crab grazers, our results suggest that shorebirds would still play a crucial role in enhancing wetland multifunctionality, for the following reasons. First, our shorebird exclusion treatments revealed that even at their current, low levels of abundances, shorebirds were still significant for enhancing wetland multifunctionality. Second, our correlational analyses showed that (i) shorebirds more strongly suppressed crab grazers at higher abundances (Supplementary Fig. 5) and (ii) wetland multifunctionality was significantly higher with fewer crab grazers (Supplementary Fig. 11). This provides additional evidence that at higher abundances, shorebirds could more strongly suppress crab grazers, leading to higher wetland multifunctionality. The consistencies between two inverse experimental treatments (excluding shorebirds and simulating high levels of shorebird predation) and between experimental and correlational evidence add to the strength of our findings.

Our findings generalize studies on the effects of shorebirds on one or a few wetland functions[49,50] and extend widely-documented effects of predators on ecosystem functions in relatively undisturbed or degraded systems[10,51] to restoration systems. Whether multifunctionality in our restoration treatments was on a trajectory to fully resemble undisturbed native wetlands is unknown due to the lack of comparable baseline sites (i.e., no wetlands with natural, healthy populations of shorebirds and crabs in this region). However, enhancing ecosystem multifunctionality may be more desirable than pursuing identical historical functions, which are often impossible due to changes in regional or global environmental backgrounds, a phenomenon known as shifting baselines[52,53].

### Harnessing the top-down effects of predators to restore ecosystem multifunctionality

Our findings have important implications for restoration of ecosystem functions. First, shorebirds are a major taxon of top predators in coastal wetlands. Our findings provide support for shorebird conservation as a potential nature-based solution to restore coastal wetlands and their multifunctionality. Shorebird conservation is often justified primarily by their endangered status and use as passive indicator species of wetland health and restoration performance[54,55]. In contrast, our findings highlight that shorebirds are critical to rebuilding the multifunctionality of coastal wetlands and suggest that shorebirds should be valued as influential drivers of ecosystem functions, even keystone species in coastal wetlands, given that they are generally ephemeral residents with small biomass relative to the marsh plants and fauna they impact[56]. Shorebird conservation may thus be more imperative than previously thought, which will require protecting and restoring nesting, foraging and breeding habitats throughout their migratory flyways, efforts that may entail eradicating invasive plants that degrade these habitats.

Furthermore, by providing experimental evidence that predators help rebuild ecosystem multifunctionality, our results support the idea of shifting ecosystem restoration from a bottom-up physical engineering perspective to a multitrophic model that integrates management of top-down trophic interactions with bottom-up restoration of foundation species. Our results suggest that continued loss of predators, if not abated, will likely limit the success of globally expanding efforts to restore ecosystems and their functions and services. Restoration exclusively focused on vegetation and other foundation species at basal trophic levels is unlikely to recover overall ecosystem functions related to trophic interactions and feedback in many ecosystems. Although the impacts of grazers might be offset through

planting at larger scales or higher densities with greater costs (the minimum planting scale or density that is necessary to allow plant establishment without managing grazers warrant further investigation), it's likely to carry considerably greater costs. Planting often costs much of the resources allowable for restoration and thus limits the potential scale of restoration[37,39], and small seedlings and other types of propagules are often highly vulnerable to grazing[41]. Harnessing the top-down effects of predators may offer a cost-effective, nature-based solution to scaling up restoration.

Harnessing the top-down effects of predators can be achieved through large-scale habitat conservation (as recognized previously; ref. 57,58), rewilding, or temporal simulation of predator effects, consumptive or non-consumptive. In ecosystems with vegetation as the foundation species, harnessing the top-down effects of predators should be focused on predators at odd-numbered trophic levels (including predators that prey on grazers) that generate positive, rather than negative, trophic cascades. A silver lining of harnessing the top-down effects of predators is that it can help optimize the design of habitat conservation so that habitats are protected and restored to allow predators to reach abundances necessary to generate their top-down effects on habitats, potentially forming a positive feedback loop that maintains ecosystems in a stable state[59]. Beyond coastal wetlands, rewilding wolves in the Yellowstone National Park has suppressed intensive elk Cervus elaphus browsing and led to increased forest productivity[60]. Rewilding sea otter populations in kelp forests has reduced sea urchin grazing and led to the recovery of kelp forests and increased finfish provision and carbon sequestration, although it decreased shellfish provision[12]. Such conflicts with certain ecosystem functions and services that society highly values have hampered recent calls for rewilding animal populations[11–13]. Our findings underscore that a multifunctionality perspective is critical for integrating top-down trophic management with bottom-up restoration.

In places where rewilding predator populations or restoring their habitat at large scales is infeasible due to socioeconomic constraints[61], our study suggests that the top-down effects of predators may be harnessed through temporal simulation of predator consumptive or non-consumptive effects. For example, despite habitat restoration at one site, shorebird populations may not recover due to failure in habitat restoration at other sites along the flyway[62]. In such cases, the top-down, consumptive effects of predators can be harnessed by excluding grazers, an approach suited mainly at small scales. But other methods, such as using baited traps to catch grazers, may be adoptable at larger scales. Also, at larger scales, simulating predators' non-consumptive effects using predator mimics, sounds, or cues may be more feasible. We suggest that developing a versatile, scalable framework for integrating top-down trophic management with bottom-up restoration is critical for enhancing the recovery of ecosystem multifunctionality and synergizing ecosystem restoration with wildlife conservation, thereby gaining broader societal support and meeting global commitments for ecosystem restoration.

## Methods
### Regional trends in shorebirds and trophic interactions
To elucidate trends in shorebird populations across the Yellow Sea, we analyzed data from multiple sources. First, we examined data from a regional shorebird survey (the Yellow Sea survey; ref. 21), where shorebird abundance was estimated at the same 14 sites using comparable methods in an early period (1996–2005) and a late period (2013–2014). These 14 sites spanned the Chinese side of the Yellow Sea from the southernmost Chongming Dongtan in the Yangtze estuary to the northernmost Liaohe estuary (see ref. 21. for detailed site information). All surveys were conducted during northward migration from late March at the southernmost site to mid-May at the northernmost site on the Yellow Sea coast, and the survey dates were selected to coincide as close as possible with the largest shorebird abundance

during northward migration at each site[21,63]. All surveys were conducted by dividing the site into several different zones and counting the number of birds on tidal flats at low or middle tides (or at pre-roosts on the upper intertidal flats or at roosting sites at high tide). Second, we examined the most up-to-date population assessment given in the database of BirdLife International (the Flyway survey; http://datazone.birdlife.org/home). To do so, we first compiled a list of shorebird species that utilize coastal habitats in the Yellow Sea based on the distribution range given by BirdLife International, and then recorded the population status (decreasing, stable, or increasing) for each species. For these two assessments of shorebird population trends in the Yellow Sea, we further categorized shorebird species into crab predators and non-crab predators based on whether or not they are known to prey on crabs[64] and counted the number of crab predators and non-crab predators with a decreasing, stable, or increasing population.

Third, to gain a more detailed understanding of shorebird population trends at our restoration site, we extracted shorebird abundance and species richness data from official annual reports of the Chongming Dongtan Bird National Nature Reserve (https://www.dongtan.cn/), covering periods (2006-2021) before and after cordgrass was eradicated in 2015. These annual surveys attempted to keep survey methods consistent over time. All surveys were conducted in late March, periods expected to have the largest shorebird abundance during northward migration. All surveys were conducted by counting the number of all bird species on tidal flats by walking along transects in four key zones of the Reserve at low or middle tides on days of spring tides. We further used data from an early survey in 1996 at the same site[63] as a baseline. Note that bird population estimates vary due to year, month, and survey method. Although ideally bird populations should be estimated using a metric of the total number of bird-days at a stopover site, such data are unavailable at our study site, as is the case for many other studies[55,65], where trends in bird populations were estimated by comparing surveys that employed similar methods. In our case, the 1996 survey was conducted also by counting all bird species in late March during northward migration but covered more zones than did the official annual bird survey. Here, to be comparable, we only considered bird numbers in the zones also covered the official annual bird survey. We only had one shorebird population estimate in 1996, which may represent an over- or underestimate of the unknown average baseline. If the 1996 estimate was an underestimate of the unknown true baseline, our statement of the shorebird population trend would be conservative. Conversely, if it's an overestimate, the shorebird population trend could be examined by comparing to the highest of the shorebird population estimates over the recent five years (2017–2021), which was still <20% of the 1996 estimate. In fact, the annual rate of shorebird population decline using the 1996 estimate as the baseline (−3.4% per year) was lower than that estimated for shorebirds with a strong reliance on the Yellow Sea habitats in a previous study (−5.2% per year; ref. 16). This suggests that our estimate of the declining trends of shorebirds was conservative. In addition, we estimated shorebird density annually by dividing abundance by the total area of tidal wetlands in these four zones in a year (where available) as previously determined via remote sensing analysis[18].

To test if shorebirds readily consume herbivorous crabs, we conducted crab-tethering experiments at five native marsh sites (spaced >1 km apart), located in the Yangtze and Yellow River estuaries. In the Yangtze Estuary, we collected *Sesarma dehaani* (carapace width 24–31 mm) from the field and placed 30 tethered crabs at each site. Tethers were constructed of 15 cm long fishing line (⌀ = 0.2 mm), tied around the carapace, and secured with cyanoacrylic glue[47]. Tethered crabs were held in place by steel stakes pushed flush with the soil surface. Note that tethering, a widely adopted approach to study predation[66–68], could overestimate predation rate. To mitigate this issue, we deployed the tethered crabs around existing crab burrows so

that crabs had access to burrows to escape shorebird predation in a relatively natural way. And we used this tethering experiment mainly to show the presence, rather than the absolute natural rate, of shorebird predation on crabs at these sites. We recorded the number of crabs that died from predation (with dismembered bodies and presence of shorebird footprints) after 24 h. We used the same procedure to quantify shorebird predation in the Yellow River estuary, except that we used another crab species (*Helice tientsinensis*) that was dominant at these sites and the number of tethered crabs was 15. Predation intensity was calculated using the equation $M_p \times 100/M$, where $M_P$ is the number of crabs that died from predation and $M$ is the total number of crabs. We used a one-sample Wilcoxon test (two-sided) to test if mean predation intensity across all sites differed significantly from zero. Throughout our analyses, data were checked for normality and homogeneity in variance, and we used nonparametric methods or generalized linear models where needed. All statistical analyses were conducted in R 4.0.3. We reported exact $P$ values where possible. Note that shorebirds may differently affect juvenile and adult crabs. Juvenile crabs are more commonly found in healthy marshes or more frequently flooded lower elevations or tidal creeks[69]. We rarely observed juvenile crabs at our study site, a restoration site at upper tidal elevations that were flooded primarily during spring tides.

To assess the effect of crab grazers on native wetland plants across the Yellow Sea, we conducted a meta-analysis. We first compiled a list of published papers by searching Scopus in December 2022 using the search string: TITLE-ABS-KEY (crab*) AND TITLE-ABS-KEY (plant* OR vegetation* OR herb*) AND TITLE-ABS-KEY (coast* OR *tidal* OR estuar* OR marsh*). We screened the resulting 1095 papers and retained those that: (1) investigated the impact of crabs on vegetation in a coastal wetland in the Yellow Sea; (2) did so using field manipulative experiments that had control and crab exclusion treatments; and (3) reported the mean values of plant biomass, standard deviation (or standard error), and sample size in both treatments. This screening yielded tests from in total seven papers (Supplementary Table 17). For each paper retained in our database, plant biomass was extracted from text and tables, or by digitizing figures using Engauge Digitizer (v 10.8). Then, we used the natural log response ratio (LRR) to evaluate the effect size in each test. LRR and associated variance (Var) were calculated as[70]:

$$LRR = \ln(X_P) - \ln(X_E) \qquad (1)$$

$$Var = \frac{S_E^2}{N_E X_E^2} + \frac{S_P^2}{N_P X_P^2} \qquad (2)$$

where $X_P$ and $X_E$ are the mean values, $N_P$ and $N_E$ are the sample sizes, and $S_P$ and $S_E$ are the standard deviations of plant biomass in control and crab exclusion treatments, respectively. Negative and positive effect sizes indicate that crabs decreased and increased plant biomass, respectively. The mean effect size was estimated using a random-effects model weighted by the inverse of variance across all tests. The mean effect size is considered significant if its 95% confidence interval (CI) does not cross zero. The random-effects model was conducted using the R metafor package.

### Restoration site

Our restoration experiment was conducted at a Wetland of International Importance (i.e., Ramsar site) in the Yangtze Estuary, Shanghai Chongming Dongtan National Nature Reserve (31°25′–31°38′ N, 121°50′–122°05′ E). The site has semi-diurnal tides with a mean tidal range of 1.96–3.08 m, a mean annual precipitation of 1022 mm, and a mean annual temperature of 15.3 °C. Soil pore water salinity ranges from 4 to 18 ppt[36,71].

The restoration site was an expansive estuarine tidal wetland that was historically dominated by *Scirpus*, a short, perennial sedge that overwinters through belowground corms and rhizomes. Exotic cordgrass was introduced to the site in the mid-1990s and spread rapidly over the past three decades, extensively replacing *Scirpus* that is currently highly endangered. In 2015, cordgrass was eradicated in tidal areas outside a man-made dike by repeatedly spraying Gallant herbicides (haloxyfop-R-methyl) which targets grasses specifically[72,73]. Cordgrass re-sprouts were monitored and eliminated by spraying the same herbicide (applications of herbicides have not been broadly applied after 2015 and have focused on small, isolated areas with cordgrass re-sprouts). The cordgrass eradication area has since been left for natural recovery (some *Scirpus* planting efforts were made outside, rather than within, the eradication area). After seven years, the cordgrass eradication area has largely remained a bare mudflat, with little (<5%) natural recovery of *Scirpus*.

The most conspicuous herbivores at the restoration site are herbivorous crabs, primarily *Sesarma dehaani* and *Helice tientsinensis*, which have been shown to rapidly graze and eliminate out-planted *Scirpus* ramets in the cordgrass eradication area[36]. Birds are mainly shorebirds including *Charadrius alexandrines*, *Calidris alpina*, *Calidris tenuirostris*, which are primary top predators in these wetlands and feed on macrofauna including herbivorous crabs. Based on the diet information reported in a local bird guidebook[64], 71% of the 48 bird species we recorded at the study site (see "Quantifying species responses and trophic interactions") are carnivores, 29% are omnivores, and none are herbivores, and 50% are known to consume crabs (see Supplementary Table 1 for a list of species that are known to consume crabs or not). Prior analysis of macrofauna showed that excluding shorebirds increased crab grazers but had little effect on other macrofauna[54]. There are no other known predators of herbivorous crabs except fish and green crabs that are few, likely due to overfishing, and that were unaffected by shorebird exclusion (see "Experimental design").

## Experimental design

The Coastal Wetland Trophic Restoration Experiment was established in the *Spartina* eradication area. Each of the four restoration treatments was randomly replicated eight times in 4 m² plots (total $n = 32$; Fig. 2). All plots were established in October 2018 and corner marked with wooden stakes. All plots had no pre-existing vegetation, similar elevation (within 5 cm), and crab burrow density (8–0 per plot). We were unable to include an undegraded native marsh reference treatment because few undegraded native marshes remained due to widespread cordgrass invasion. We focused on examining if restoration treatments could enhance ecosystem multifunctionality compared to natural recovery control. As the site has been degraded for more than a decade without a persistent bank of *Scirpus* propagules (seeds and corms), planting (or seeding) is now a necessary condition for vegetation recovery[74]. We thus omitted a natural recovery + crab exclusion treatment. All plots were monitored every 1–2 weeks (except during the COVID-19 lockdown periods in February–March 2020), and treatments were maintained as necessary.

For all plots assigned planting treatments, we planted nine clumps of *Scirpus* each containing 15 ramets of a similar size (15–20 cm tall; clumps were arranged by 3 × 3, with even spacing). Planting was conducted in May 2019, and *Scirpus* clumps were excavated from a nearby vegetated area.

Shorebird exclusion treatments were established by corner marking plots with wooden stakes and attaching five lines of dark green nylon strings tied to the corner stakes around the periphery to exclude shorebirds from entering the plots. The bottom and top lines were ~5 cm and 120 cm above the soil surface with other lines spaced at ~10–30 cm intervals. The same nylon strings (arranged by 3 × 3, with even spacing) were used to cover the top of the plots to prevent birds

from flying into the plots. This design effectively excluded shorebirds while allowing few other predators such as fish and green crabs to move freely. The number of shorebird footprints was reduced by >80% in these plots[54]. Given the large spaces (ca.10–30 cm) in between, the nylon strings used to exclude shorebirds, per se, had negligible impacts on physical factors such as light and water flow. Similar bird exclusion methods have been used previously[75,76].

For high predation simulation treatments, we constructed exclusion cages of nylon mesh (1 cm mesh size) attached to a PVC frame (the nylon mesh we used did not significantly affect physical factors such as light or water flow in previous work where we quantified wave attenuation in high predation simulation and natural recovery treatments during a winter period when all treatments had no existing plant canopies[54]). Crab exclusion cages were roofless, 0.5-m high aboveground, and extended 0.3 m deep belowground to prevent crabs from entering the plots by burrowing. Crab grazers were removed from these plots by installing seven PVC pitfall traps (10 cm diameter, 20 cm depth) flush with the surface in each plot. Crabs that fell into these traps were removed every 1–2 weeks. The crab removal treatments were maintained almost free of grazing crabs for over three years to simulate high shorebird predation that occurred historically before the collapse of the migratory shorebird flyway or that could occur with shorebird recovery. This simulation was warranted, for the following reasons:

(i) Even under current, low levels of shorebird abundance, shorebirds suppressed the abundance of crab grazers at our study site by on average 68%, and by up to 90% during seasons when shorebird abundance was high (see Fig. 2b). (ii) Assuming that shorebirds suppressed crab population in proportion to their abundance, the abundance of crab grazers ($D_{1996}$) at the study site before shorebird population collapse in 1996 could be estimated using the equation:

$$D_{1996} = D_{C2021} + [(D_{C2021} - D_{E2021})/(p \times N_{2021})] \times (N_{1996} - N_{2021}) \quad (3)$$

where $D_{C2021}$ and $D_{E2021}$ are the means of crab grazer abundance in control and shorebird exclusion treatments, respectively, during northward migration in 2021, $p$ the mean proportional change in shorebird abundance (number of footprints) between control and shorebird exclusion treatments during northward migration in 2021 (ref. 54), and $N_{1996}$ and $N_{2021}$ the shorebird population during northward migration in 1996 and 2021, respectively. This calculation yielded a negative value, suggesting potential elimination of crab grazers by shorebirds of historical abundances. (iii) By simulating the non-consumptive effects of shorebirds (detailed in the following section), we found that the presence of a swinging shorebird model (i.e., a predation cue[77]) reduced crab abundance to almost nil over just a few weeks (Supplementary Fig. 10), further confirming that even in the absence of consumptive effects, shorebirds can nearly eliminate crab grazers. (iv) Crab grazers were nearly absent at sites where shorebirds were most abundant in the Yellow Sea[24]. (v) At high abundances, other predators have also been shown in many studies to drive prey species to become rare, even locally extinct[25–32].

## Quantifying species responses and trophic interactions

To quantify the abundance and species richness of shorebirds at the restoration site, we used infrared camera-traps, which have been increasingly used to estimate bird populations with minimum observer interference[78]. We mounted three cameras on wooden poles 1.5 m above the ground in March 2019. Cameras were separated by >50 m and covered a field of view of ~0.8 ha in total. The cameras were set to take photographs at 5-min intervals from 2019 through 2021. We identified each bird species (when possible) and counted their number in each photograph, and then summed up all bird species and their abundance by month as a measure of bird species richness and abundance, respectively. We categorized all recorded birds into

shorebirds and non-shorebirds (i.e., all birds other than shorebirds), which were further categorized into crab predators and non-crab predators, respectively, based on the diet information reported in a local bird guidebook[64] (see Supplementary Table 1).

To test if the shorebird exclusion treatment decreased shorebird predation on crab grazers, we conducted a crab-tethering experiment using the same tethering method described above in regional trends in shorebirds and trophic interactions. We tethered and placed two *Sesarma dehaani*s in each plot (except the plots where grazing crabs were excluded to simulate high levels of shorebird predation). We recorded the number of crabs that died from predation in each plot every day for a week. Differences in shorebird predation intensity between treatments were analyzed with a pairwise proportion test (two-sided), and *P* values were adjusted with the Bonferroni method.

To quantify crab abundance in each treatment, we counted the number of crab burrows in the center ($1 \times 1$ m) of each plot every 1–2 weeks (we omitted crab exclusion plots where grazing crabs were actively removed using pitfall traps). Crab burrow density has been widely used as a proxy of crab abundance[79,80]. We calculated the monthly average crab burrow density in each plot, $x + 1$ transformed the data to avoid non-positive values, and analyzed the effects of experimental treatments, time, and their interaction using a generalized linear mixed model (GLMM, Gamma distribution) with plot ID as a random effect. GLMM was conducted with the lme4 package (version 1.1.34) in R.

To examine the effect of shorebirds on crab abundance in our manipulative experiment, we first calculated an effect size by using Eqs. 1 and 2, but here $X_P$ and $X_E$ are the mean values, $N_P$ and $N_E$ the sample sizes, $S_P$ and $S_E$ the standard deviations of crab abundance in planting + shorebird control and planting + shorebird exclusion treatments, respectively. We then constructed meta-regression models using the R metafor package (version 4.4.0) to test whether the effect sizes varied with the monthly total abundance and species richness of all birds. We reran this analysis using the monthly abundances of all shorebirds (i.e., Charadriiformes), all non-shorebirds, crab predators and non-crab predators of shorebirds and non-shorebirds, six shorebird species and two non-shorebird species with the highest total abundance, respectively. In all these analyses, time (month/year) was included as a random effect.

To quantify *Scirpus* recovery in each treatment, we counted *Scirpus* shoots in each plot every two days for the first two weeks since planting and then every 1–2 weeks during the growing season of each year (April through September). We estimated the initial rates of plant abundance change over the first week in each plot using a linear regression. Natural recovery treatments were omitted from analyses because they had no plants throughout the experiment. We used the agricolae package (version 1.3.7) in R to test for differences in the rate of plant abundance change among experimental treatments with a one-way ANOVA, followed by Tukey's HSD multiple comparisons. We further analyzed the effects of all experimental treatments, time, their interaction on plant abundance at the end of each month using a GLMM (Poisson distribution) with plot ID as a random effect.

Additionally, to investigate if shorebirds had non-consumptive effects on crab grazers, we conducted an experiment where potential non-consumptive effects of shorebirds were simulated by using a shorebird model. The experiment included three treatments each replicated eight times in $2 \times 2$ m plots at the restoration site: (i) shorebird model: a swinging shorebird (an adult *Numenius phaeopus*) model was installed to simulate the non-consumptive effects of shorebirds. The shorebird model was held -1.5 aboveground with strings tied to two wooden stakes on the periphery of a plot, and the shorebird model was of a flying mode and could swing with wind to simulate the flying behavior of shorebirds; (ii) procedural control: all treatments were the same as shorebird model plots, except that we used a ball with similar size and color, instead of a shorebird model, to

test if the presence of the same materials but without the visual cues of a shorebird can induce similar responses of crabs; and (iii) control: no additional treatments except that the plot was marked at four corners (Supplementary Fig. 10). We counted the number of crab burrows in the center ($1 \times 1$ m) of each plot two days before, on, and then weekly after the day (June 20) when all treatments were set up. We tested for differences in crab abundance among treatments for each date using a one-way ANOVA, followed by Tukey's HSD multiple comparisons.

### Quantifying ecosystem functions

The 12 ecosystem functions were quantified in each of the 32 experimental plots in 2021, using established methods as described below.

**Biological functions.** To quantify primary production, we used end of growing season *Scirpus* biomass as a proxy (no other plants existed in our plots). In each plot, we harvested a total of 30 plants from within three haphazardly placed quadrats ($50 \times 50$ cm; 10 plants per subplot). Each plant was then measured for stem height and dried at 65 °C until constant weight. An allometric equation of *Scirpus* aboveground biomass against stem height was then constructed: $M = 0.0004 \times H^{1.72}$ ($R^2 = 0.87$), where $M$ and $H$ are the aboveground biomass and height of *Scirpus*, respectively. The total aboveground biomass of all *Scirpus* ramets in a plot was estimated by calculating the aboveground biomass of a *Scirpus* ramet based on the average ramet height in that plot and multiplying it with *Scirpus* density per unit area. To determine *Scirpus* belowground biomass, triplicate soil cores (11 cm diameter, 20 cm depth) were haphazardly taken from each plot and sieved through 1 mm mesh. *Scirpus* belowground biomass per soil core was then dried at 65 °C until constant weight. Total belowground biomass of *Scirpus* per plot was estimated by averaging and standardizing the belowground biomass from the three soil cores of a plot to unit area (per m²). Primary production was estimated as the sum of *Scirpus* above- and belowground biomass. We omitted algal production that was negligible (our site often completely dried up during neap tide periods and had no macroalgae).

To quantify secondary production, we used the wet biomass of macrofauna as a proxy. Three soil cores (11 cm diameter, 20 cm depth) were taken from each plot in September 2021. Soil cores from each plot were combined and sieved through 1 mm mesh, and the retained organisms were preserved in 75% ethanol. Macrofauna were then identified to the finest taxonomic resolution as possible, counted, and weighed by species or taxonomic group. Secondary production was standardized to unit area (g wet biomass m⁻²) and macrofaunal species richness was expressed as the number of species.

To quantify microbial production, we employed the real-time quantitative PCR (qPCR) method[81]. Three soil cores (2.5 cm diameter, 15 cm depth) were taken with a stainless-steel corer at random locations in each plot, mixed as one composite sample, and brought back to the laboratory on dry ice and kept at −80 °C for further analysis. Each sample was then sent for DNA extraction and quantitative PCR (by Shanghai Majorbio Bio-pharm Technology Co., Ltd; see Supplementary Table 18 for the detailed procedure). Microbial production was expressed as gene copies g⁻¹ soil. See Appendix S1 for detailed methods for quantifying the operational taxonomic units (OTUs) of microbes.

**Physical functions.** To quantify wave attenuation, we used gypsum dissolution blocks that dissolve at a rate proportional to water velocity, and thus are an integrated proxy for hydrodynamic flow[82]. We made gypsum dissolution blocks as hemispheres ($\varnothing = 6.5$ cm) from dental plaster and covered the bottom with two layers of polyurethane so that the surface area subject to dissolution was equal[34,83]. The initial mass of gypsum blocks was weighed after being dried at 60 °C for 24 h. One gypsum dissolution block was deployed flush with the soil surface in each plot in August, retrieved after 14 d, dried, and reweighed. The rate

of dissolution was calculated as grams of gypsum dissolved per day (Supplementary Fig. 16, Supplementary Table 19). To ensure that all greater function values indicate greater benefits in view of ecosystem services, wave attenuation was estimated by using the equation $-f_i + \max(f_i)$, where $f_i$ is the value of dissolution rate in plot $i$ and $\max(f_i)$ is the average of the three maximum values across all plots[3,84,85].

To quantify marsh infiltration, we used a double-ring infiltrometer[4,33]. We placed a 1.5 L PVC cylinder (11 cm diameter, 16 cm depth) in each plot in October 2021, filled it with 1 L of creek water at low tide, and determined the time required for the water to drain out of the ring. Water infiltration was expressed as water drainage in liters per hour.

To quantify sediment accretion, we inserted three 1.5-m long PVC poles vertically into the sediment at three randomly chosen locations in each plot until resistance[4,34]. Each pole was marked 20 cm above the substrate surface in April 2021, and the distance between the mark and the substrate surface was re-measured monthly from April to October 2021. We calculated sedimentation accretion rate as the difference between the first and last measurements divided by the number of months (cm mo$^{-1}$).

**Biogeochemical functions.** To quantify soil respiration, we installed a PVC collar (20 cm diameter, 20 cm depth) in each plot. Soil respiration was then measured on sunny days during 9:00-11:30 at neap tide in October 2021, using a portable LI-8100A soil $CO_2$ flux system (LICOR, Lincoln, Nebraska, USA). Prior to measuring soil respiration, any aboveground plant material within the PVC collar was removed. Three measurements (in μmol $CO_2$ m$^{-2}$ s$^{-1}$) were taken per plot and averaged for analysis.

To quantify soil nitrogen mineralization, we collected six soil cores (7 cm diameter, 15 cm depth) from each plot in October 2021, three of which were combined as a composite sample for determining soil nitrate ($NO_3^+$-N) and ammonium ($NH_4^+$-N) concentrations and summed as total inorganic nitrogen ($SIN_{t0}$). Meanwhile, the other three soil cores were put into PVC tubes (7 cm diameter, 15 cm depth) with lids sealing the bottom and polytetrafluoroethylene (PTFE) films covering the top (allowing only gas to be exchanged with the soil in the tube) and incubated in situ for 30 days. Then the soil cores from each plot were retrieved and combined as a composite sample for determining soil nitrate and ammonium concentrations, which further summed as total inorganic nitrogen ($SIN_{t30}$) in the lab. To determine soil nitrate and ammonium concentrations, 20 grams of wet soil was added to 200 mL of 2 M KCl solution., shaken at 200 rpm for 1 h, centrifuged and filtered to obtain an extract. Extracted $NH_4^+$-N and $NO_3^+$-N were analyzed using the indophenol blue method and dual-wavelength method, respectively[86,87]. Net mineralization rate (mg kg$^{-1}$ day$^{-1}$) was calculated as changes in soil inorganic nitrogen per day.

To quantify litter decomposition, standing dead *Scirpus* stems were collected from a vegetated area, washed, and dried at 65 °C until constant weight. Ten grams of stems were placed in each of 96 mesh bags (15 × 20 cm, 1 mm mesh), three of which were staked on the sediment surface in each plot in August. After 1, 2, and 3 months, one of the three mesh bags in each plot was retrieved and the remaining material was washed, dried, and weighed. An exponential equation fitting *Scirpus* mass remaining against time was then constructed: $L_t = L_0\, e^{-kt}$, where $L_0$ and $L_t$ are litter mass at time 0 and $t$ (mo), respectively, and $k$ the decomposition rate (mo$^{-1}$).

To quantify sediment carbon burial, three soil cores (2.5 cm diameter, 15 cm depth) were taken per plot in September 2021, mixed into one composite sample, and transported to laboratory in ziplock bags. Soils were air-dried and sieved through a 0.1-mm mesh after removing plant roots and rhizomes. To remove carbonates, we acidified 10 g of air-dried soil samples using 1 mol L$^{-1}$ HCl[87]. Acidified samples were oven dried, grounded, sieved through 0.1-mm mesh, and analyzed for carbon content (i.e., soil organic carbon concentration, %) using an element analyzer (vario MACRO cube, Elementar, Germany). To measure soil bulk density, we additionally collected one intact soil core (100 cm$^3$) per plot, which was dried and weighed in the lab. Bulk density (g cm$^{-3}$) was calculated using the equation $M_d / V$, where $M_d$ is the mass of dry soil and $V$ is the volume of soil core. Sediment carbon burial rate (g C m$^{-2}$ mo$^{-1}$) was estimated by multiplying sediment accretion rate by carbon density, where sediment accretion rate was determined previously and carbon density was calculated by multiplying soil organic carbon concentration (%) by soil bulk density.

To assess nitrogen accumulation in soil, unacidified soil samples used to determine sediment carbon burial were packeted in tin foil and analyzed for soil total nitrogen using the same element analyzer as described above for carbon. Soil nitrogen accumulation (g N m$^{-2}$ mo$^{-1}$) was estimated similarly as described above for carbon sequestration.

To quantify soil heavy metal reduction, 0.5 g unacidified soil samples collected for determining sediment carbon burial were placed in 50 ml digestion tubes, soaked, and digested with 8 mL of a digestant mixture (HCl: HNO₃, v-v = 1:1) for 8–10 h at 140–160 °C. The digested samples were then tested for concentration of seven heavy metals (Cadmium, Arsenic, Lead, Zinc, Chromium, Copper, and Nickel) using plasma mass spectrometry (ICP-MS; Elan DRCe, PerkinElmer). To obtain a measure of soil heavy metal pollution, we calculated the Nemerow multifactor index for each plot (see Appendix S2 for detailed methods). Soil heavy metal reduction was estimated by converting the Nemerow multifactor index using the same method as given for wave attenuation.

## Ecosystem function analyses

To assess the effects of restoration treatments on ecosystem functions individually, we used one-way ANOVAs, followed by Tukey's HSD multiple comparisons when one-way ANOVAs yielded significant differences ($P < 0.05$). A nonparametric Wilcoxon test (two-sided) followed by pairwise comparisons (adjusted with the bonferroni method) was used for primary production and secondary production that did not meet the assumptions of ANOVAs.

To assess the effects of restoration treatments on multifunctionality, we used two different approaches. Prior to analysis, all data on an ecosystem function were standardized as percent of the maximum observed value of that function[3]. First, we calculated an unweighted, mean multifunctionality index for each plot. We used an unweighted approach so that all ecosystem functions were treated the same, following previous studies[34,85]. A one-way ANOVA was used to test for the effects of restoration treatments on the average multifunctionality index, followed by Tukey's HSD multiple comparisons. This analysis on the average multifunctionality index was conducted for all 12 functions and then for above- and belowground functions separately. Aboveground functions included aboveground biomass, wave dissipation, sediment accretion, and litter decomposition, while belowground functions included belowground biomass, secondary production, microbial production, marsh infiltration, soil respiration, nitrogen mineralization, sediment carbon burial, nitrogen accumulation, and soil heavy metal reduction. Note that although we omitted biodiversity functions in our analysis (e.g., macrofaunal species richness or the microbial OTUs; see Supplementary Fig. 17, Supplementary Table 20), substituting secondary production with macrofaunal species richness, and microbial production with the number of microbial OTUs did not alter our results on multifunctionality (Supplementary Fig. 18, Supplementary Table 21).

Second, as the average multifunctionality index cannot indicate whether all functions perform at a high level simultaneously (given that functions performing at high levels can be averaged out by those performing at low levels[34]), we analyzed multifunctionality using the effective number of functions (which is analogous to the effective number of species used to quantify species diversity) and effective multifunctionality, a measure of the cumulative performance of the

system were all functions provided equally[88]. To compute the effective number of functions, we standardized the 12 measured functions to a common scale of 0–1, where 0 means no function and 1 means the maximum level of a function. The effective number of functions can be computed using the following equations:

$$N^q = \left( \sum_{i=1}^{k} p_i^q \right)^{1/(1-q)} \quad q \geq 0, q \neq 1 \quad (4)$$

$$p_i = \frac{F_i}{\sum F_i} \quad (5)$$

where $p_i$ is the relative proportion a function ($p_i$) contributes to the whole, $F_i$ is the standardized value of function $i$ ($i = 1, 2, … k$), and $N^q$ is the effective number of functions for order $q$, a factor for how sensitive the function should be to differences in the level at which functions are provided[88]. Larger values of $q$ express the degree to which high-performing functions are upweighted. Equation 4 is undefined for $q = 1$, but its limit $q \to 1$ is:

$$N^1 = \lim_{n \to 1} N^q = \exp \left( - \sum_{i}^{k} P_i \times \log P_i \right) \quad (6)$$

The effective number of functions is just the number of functions measured when order $q = 0$ and analog to Shannon diversity for species when $q = 1$. When $q > 1$, functions performing at higher levels are given greater weight. For each plot, we calculated the effective number of functions with $q$ from 1 to 5. We further computed the effective multifunctionality of order $q$ ($^qM_{ef}$) as:

$$^qM_{ef} = N^q \times A \quad (7)$$

where $A$ is the arithmetic mean of the standardized values of the 12 measured functions. A one-way ANOVA was then used to test for the effects of restoration treatments on the effective number of functions and effective multifunctionality for each order of $q$, followed by Tukey's HSD multiple comparisons.

Additionally, we examined the relationships between functions and primary productivity and crab abundance, and among different functions. To examine if primary productivity was indicative of the 11 other ecosystem functions and the mean multifunctionality index, we conducted linear and quadratic regressions with the bbmle package (version 1.0.25) in R, and selected the best fitting model based on Akaike information criterion corrected for small sample size. We similarly conducted linear and quadratic regressions to examine if the abundance of grazing crabs (the mean annual value of crab abundance per plot in 2021) was indicative of the 12 ecosystem functions individually and collectively. We further calculated Spearman correlations among all 12 functions.

### Reporting summary
Further information on research design is available in the Nature Portfolio Reporting Summary linked to this article.

## Data availability
The survey data of shorebird abundance at 14 sites in the Yellow Sea are available in a previous publication (ref. [21]). Assessments of the global population trends of shorebird species that utilize coastal habitats in the Yellow Sea are available in the database of BirdLife International (http://datazone.birdlife.org/home). Soil microbial sequences have been deposited in the NCBI Sequence Read Archive under project number SRP472937. All other data needed to evaluate the conclusions in the paper have been deposited in the public repository Zenodo (https://doi.org/10.5281/zenodo.10172655) (ref. [89]).

## Code availability
Code used to analyze the data of this study has been deposited in the public repository Zenodo (https://doi.org/10.5281/zenodo.10172655) (ref. [89]).

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

## Acknowledgements

We thank Shuijin Hu, Bo Li, Zhijun Ma, and Jihua Wu for advice on this study. We thank the many student volunteers who helped with identifying and counting bird species. Access to the study sites in the Yangtze Estuary and the Yellow River Estuary was granted by the Management Offices of Shanghai Chongming Dongtan National Nature Reserve and the Shandong Yellow River Delta National Nature Reserve, respectively. This work was funded by National Natural Science Foundation of China (grant no. 32271601 and 31870414 to Q.H.) and Shanghai Basic Research Special Zone Program (grant no. 22TQ011 to Q.H.).

## Author contributions

Q.H. designed the project; C.L. and J.C. conducted the exepriment and collected the data with the help from X.L.; C.L. and Q.H. analyzed and visualized the data; C.L. and Q.H. wrote the first draft; and Q.H., C.L., J.C., X.L., A.P.R., C.A., L.L., B.R.S. and M.D.B. contributed substantially to reviewing and editing the manuscript.

## Competing interests

The authors declare no competing interests.
