## [Peer Review File · Nature Communications]

Shorebirds-driven trophic cascade helps restore coastal wetland multifunctionalityEditorial Note: This manuscript has been previously reviewed at another journal that is not operating a transparent peer review scheme. This document only contains reviewer comments and rebuttal letters for versions considered at Nature Communications.

REVIEWERS' COMMENTS

Reviewer #2 (Remarks to the Author):

The authors have done an excellent job in adequately addressing all my comments on the previous version. These mainly concerned how they addressed the multifunctionality analyses. It also seems they have responded well to the issues raised by the other reviewers.

Minor comments:

Fig. 3. In (A) red colour indicates all shorebirds, which means high predation. In (B) to (F) however, red indicates shorebird exclusion meaning low predation. I encourage the authors to revise the colors to be consistent here. Also, please be consistent with the order of the treatments in insets (C) and (F).

Supplementary Table 3: Please also include authors.

Reviewer #4 (Remarks to the Author):

The authors have done a great job in revising their manuscript by addressing most of the comments and suggestions raised by reviewers. The manuscript has been greatly improved, with more balanced and comprehensive discussions and suggestions. I have a few additional comments, some of which related to the points made by other reviewers in the first round and the authors' replies, which may need to be addressed to make the authors' suggestions for habitat restoration and shorebird conservation clearer and more logical.

1. In the authors' field experiment, the planting of native *Scirpus* was conducted at relatively small scale compared with the expansive tidal flat. As a result, in the planting + shorebird control group, where crabs are allowed to graze on newly planted *Scirpus*, it is possible that crabs from a much larger area are drawn to the plots, thus may overestimate the crabs' effect on *Scirpus* (re)establishment. In other words, if *Scirpus* planting was conducted over larger areas, say, converting 10% of the historical/potential suitable tidal flat at a site, or at the scale of habitat restoration projects, will crabs still prevent the establishment of *Scirpus*?

2. Related to the comment above, there is another layer in the scalability issue: if *Scirpus* planting can be or will be conducted over larger areas in a restoration effort, can the authors elaborate or speculate what would or could be the proper size that may see positive establishment of *Scirpus*? How does the effort in restoring larger size of *Scirpus* compare with that in reducing crab abundance? Which might be the most cost-effective solution to scale up *Scirpus* restoration?

3. The authors' suggestion that top-down effect could facilitate habitat restoration or ecosystem multifunctions through reestablishing vegetation may only applied to systems with three focal trophic levels, while in systems where top-down effects involve two or four trophic levels, the introduction or conservation of top predators may reduce native vegetation, with potential negative effect on habitat structure and functionality (although it depends on the baselines). Could the authors add a short discussion addressing this point as to how the number of trophic levels in a trophic cascade may (or may not) render top predator conservation effective/beneficial?

4. In the authors' reply to multiple reviewers' comments, they mentioned that egrets and herons have been shown/observed to have major effect on crab abundance, while the abundance of egrets and herons in the study areas was low. However, in their supplementary figure 6, Ardeidae spp. were included and appeared to have very high abundance, if I read the figure correctly. The statement and figure seem to be conflicting.

Reply to Reviewer Comments

Reviewer Comments:

Reviewer #2 (Remarks to the Author):

The authors have done an excellent job in adequately addressing all my comments on the previous version. These mainly concerned how they addressed the multifunctionality analyses. It also seems they have responded well to the issues raised by the other reviewers.

Reply: We thank the reviewer for reviewing our revised manuscript and for their positive comments on our revised manuscript. We have carefully considered their additional comments and revised the manuscript accordingly. Please see our point-to-point response to their comments and our revisions detailed below.

Minor comments:

Fig. 3. In (A) red colour indicates all shorebirds, which means high predation. In (B) to (F) however, red indicates shorebird exclusion meaning low predation. I encourage the authors to revise the colors to be consistent here. Also, please be consistent with the order of the treatments in insets (C) and (F).

Reply: Following the reviewer's suggestion, we revised the color in b to f so that red colors indicate high predation treatments. We also confirmed that the treatments in insets c and f are ordered consistently. Note that the planting + high predation simulation treatment is absent in c, as we excluded crabs in this treatment and that the natural recovery + shorebird control treatment is absent in f, as this treatment had no plants throughout the experiment (clarified on line 994 in the revised manuscript).

Supplementary Table 3: Please also include authors.

Reply: Authors are now included (Supplementary Table 17 in the revised manuscript).

Reviewer #4 (Remarks to the Author):

The authors have done a great job in revising their manuscript by addressing most of the comments and suggestions raised by reviewers. The manuscript has been greatly improved, with more balanced and comprehensive discussions and suggestions. I have a few additional comments, some of which related to the points made by other reviewers in the first round and the authors' replies, which may need to be addressed to make the authors' suggestions for habitat restoration and shorebird conservation clearer and more logical.

Reply: We thank the reviewer for reviewing our revised manuscript and for the positive comments. We have carefully considered their additional comments and revised the manuscript accordingly. Please see our point-to-point response to their comments and our revisions detailed below.

1. In the authors' field experiment, the planting of native Scirpus was conducted at relatively small scale

compared with the expansive tidal flat. As a result, in the planting + shorebird control group, where crabs are allowed to graze on newly planted Scirpus, it is possible that crabs from a much larger area are drawn to the plots, thus may overestimate the crabs' effect on Scirpus (re)establishment. In other words, if Scirpus planting was conducted over larger areas, say, converting 10% of the historical/potential suitable tidal flat at a site, or at the scale of habitat restoration projects, will crabs still prevent the establishment of Scirpus?

Reply: In the revised manuscript, we have included additional discussions to clarify the relevance of our findings to real-world restoration projects.

First, we have now acknowledged that some crabs nearby might have been attracted to plantings at small plot scales, which could contribute to the observed effects of crab grazing. However, we have observed that at real-world restoration projects conducted at our study site and in other sites in the Yellow Sea (He *et al.* 2017), crabs still intensively constrained the establishment of marsh plantings. This is consistent with a recent global synthesis that shows that the suppressing effects of herbivores on plant abundance at restoration sites often did not vary significantly with plot size and remained strong in restoration studies that used > 1 ha plots (Xu *et al.* 2023), which are larger than the sizes of many restoration sites (Bayraktarov *et al.* 2016). These are clarified in the revised manuscript (lines 151-157).

Second, even if the effects of grazers can be offset through planting at large scales or higher densities, which would carry greater costs, harnessing the top-down effects of predators (such as reducing crab abundance by using shorebird mimics or crab traps) may lower the efforts and costs of planting that is necessary to ensure success and thus offer a potentially cost-effective, nature-based solution to scaling up restoration (discussed on lines 299-306 in the revised manuscript).

2. Related to the comment above, there is another layer in the scalability issue: if Scirpus planting can be or will be conducted over larger areas in a restoration effort, can the authors elaborate or speculate what would or could be the proper size that may see positive establishment of Scirpus? How does the effort in restoring larger size of Scirpus compare with that in reducing crab abundance? Which might be the most cost-effective solution to scale up Scirpus restoration?

Reply: In addition to our response to the reviewer's comment above, we have also discussed that even if the effects of crabs can be offset through planting at large scales or higher densities (the specific planting scale or density that is necessary to allow the establishment of *Scirpus* without reducing crab grazers warrant further investigation), it's likely to carry considerably greater costs. Planting often costs much of the resources allowable for restoration and thus limits the potential scale of restoration (Bayraktarov *et al.* 2016, Chazdon & Guariguata 2016), and small plant seedlings and propagules are often highly vulnerable to grazing (Xu *et al.* 2023). Harnessing the top-down effects of predators (such as reducing crab abundance by using shorebird mimics or crab traps) may offer a potentially cost-effective, nature-based solution to scaling up restoration. See lines 299-306 in the revised manuscript.

3. The authors' suggestion that top-down effect could facilitate habitat restoration or ecosystem multifunctions through reestablishing vegetation may only applied to systems with three focal trophic levels, while in systems where top-down effects involve two or four trophic levels, the introduction or conservation of top predators may reduce native vegetation, with potential negative effect on habitat structure and functionality (although it depends on the baselines). Could the authors add a short discussion addressing

this point as to how the number of trophic levels in a trophic cascade may (or may not) render top predator conservation effective/beneficial?

Reply: In the revised manuscript (lines 309-312), we have added discussions on how the number of trophic levels in a trophic cascade may render top predator conservation effective/beneficial. Predators at odd-numbered, rather than even-numbered, trophic levels may be beneficial for restoring vegetation and associated ecosystem functions.

4. In the authors' reply to multiple reviewers' comments, they mentioned that egrets and herons have been shown/observed to have major effect on crab abundance, while the abundance of egrets and herons in the study areas was low. However, in their supplementary figure 6, Ardeidae spp. were included and appeared to have very high abundance, if I read the figure correctly. The statement and figure seem to be conflicting.

Reply: We meant that the abundance of non-shorebirds including egrets and herons at our study site was low and did not significantly affect crab abundance (lines 172-175 in the revised manuscript). The Supplementary Fig. 7 only showed the abundances of non-shorebirds (shorebirds were excluded from this figure). Although *Ardeidae* spp. had the highest abundance among non-shorebirds, its abundance was still low compared with shorebirds (Fig. 3a and Supplementary Figure 4). To avoid confusion, we have now rescaled the y axis of Supplementary Fig. 7 so that it is consistent with Fig. 3a and Supplementary Figure 4.

References

- Bayraktarov, E. *et al.* The cost and feasibility of marine coastal restoration. *Ecol. Appl.* 26, 1055-1074 (2016).
- Chazdon, R. L. & Guariguata, M. R. Natural regeneration as a tool for large-scale forest restoration in the tropics: Prospects and challenges. *Biotropica* 48, 716-730 (2016).
- He, Q., Silliman, B. R., Liu, Z. & Cui, B. Natural enemies govern ecosystem resilience in the face of extreme droughts. *Ecol. Lett.* **20**, 194-201 (2017).
- Xu, C. *et al.* Herbivory limits success of vegetation restoration globally. *Science* **382**, 589-594 (2023).